# Distribution of Presepsin, Krebs von den Lungen 6, and Surfactant Protein A in Umbilical Cord Blood

**DOI:** 10.3390/diagnostics12092213

**Published:** 2022-09-13

**Authors:** Minjeong Nam, Mina Hur, Hanah Kim, Gun-Hyuk Lee, Mikyoung Park, Han-Sung Kwon, Han-Sung Hwang, In-Sook Sohn

**Affiliations:** 1Department of Laboratory Medicine, Korea University Anam Hospital, Seoul 02841, Korea; 2Department of Laboratory Medicine, Konkuk University School of Medicine, Seoul 05030, Korea; 3Department of Laboratory Medicine, Eunpyeong St. Mary’s Hospital, Seoul 03312, Korea; 4Department of Obstetrics & Gynecology, Konkuk University School of Medicine, Seoul 05030, Korea

**Keywords:** presepsin, KL-6, SP-A, reference interval, umbilical cord blood

## Abstract

Presepsin is an early indicator of infection, and Krebs von den Lungen 6 (KL-6) and Surfactant Protein A (SP-A) are related to the pathogenesis of pulmonary infection and fibrosis. This study aimed to establish reference intervals (RIs) of presepsin, KL-6, and SP-A levels and to evaluate the possible influence of neonatal and maternal factors on presepsin, KL-6, and SP-A levels in umbilical cord blood (UCB). Among a total of 613 UCB samples, the outliers were removed. The RIs for presepsin, KL-6, and SP-A levels were defined using non-parametric percentile methods according to the Clinical and Laboratory Standards Institute guidelines (EP28-A3C). These levels were analyzed according to neonatal and maternal factors: neonatal sex, gestational age (GA), birth weight (BW), Apgar score, delivery mode, the presence of premature rupture of membranes (PROM), gestational diabetes mellitus (GDM), and pre-eclampsia. Presepsin, KL-6, and SP-A levels showed non-parametric distributions and left-skewed histograms. The RIs of presepsin, KL-6, and SP-A levels were 64.9–428.3 pg/mL, 43.0–172.0 U/mL, and 2.1–36.1 ng/mL, respectively. Presepsin, KL-6, and SP-A levels did not show significant differences according to sex, GA, BW, Apgar score, delivery mode, PROM, GDM, and pre-eclampsia. The median level and 97.5th centile RI of KL-6 showed a slight increase with increased GA. We established RIs for presepsin, KL-6, and SP-A levels in large-scaled UCB samples. Further investigation would be needed to determine the clinical significance.

## 1. Introduction

Neonatal sepsis and lower respiratory infection are some of the leading causes of neonatal morbidity and mortality [1]. Neonatal sepsis is a syndrome characterized by non-specific signs and symptoms of systemic infection [2]. Neonatal respiratory infection is a heterogeneous syndrome with diverse and unknown etiology [3]. To be diagnosed with neonatal sepsis or respiratory infection, a physical examination is the cornerstone of clinical practice, and a blood culture or a chest X-ray is a gold standard [3,4]. However, clinical manifestations in neonates are ambiguous and non-specific. The conventional blood culture takes a long turn-around time and shows a high false-negative rate because of inadequate blood volume [5]. Exposure to radiation in neonates may increase the risk of cancer [6].

Given the limitations of the clinical manifestations and current gold standard, an accurate biomarker is necessary for the timely and accurate diagnosis of neonatal sepsis and respiratory infection. For neonatal sepsis and respiratory infection, there are many biomarkers commonly used in clinical practice: C-reactive protein (CRP), procalcitonin, interleukins, and cell adhesion molecules [4,7]. However, no single biomarker is relevant to others in diagnosing neonatal sepsis or respiratory infection.

A cluster of differentiation 14 (CD14) is a glycoprotein expressed in the membrane surface of diverse cells and serves as a high-affinity binding site to lipopolysaccharides (LPSs); it is implicated in the recognition of various bacterial products [8]. Presepsin is the truncated form of the soluble CD14 (sCD14). Presepsin is a new biomarker with clinical utility in early inflammation and sepsis, proven to have high sensitivity and specificity [9,10,11,12]. Krebs von den Lungen 6 (KL-6) and Surfactant Protein A (SP-A) are expressed in type II alveolar epithelial cells and are associated with lung injury [13]. The levels of KL-6 and SP-A are increased in interstitial pneumonia [14].

Reference intervals (RIs) are the most widely used clinical decision-making tools that help clinicians diagnose diseases. The RIs could vary according to sex, age, and ethnicity [15]. Thus, each clinical laboratory should validate the RIs in a local population [16]. The validation of RIs for neonates is very difficult because of the limitation of obtaining sufficient reference neonatal samples [17]. Alternatively, umbilical cord blood (UCB) is collected at birth without any invasive procedure [18]. Thus, UCB could be a good alternative in neonatal studies, which have many restrictions.

Although the previous studies investigated the diagnostic value for presepsin, KL-6, and SP-A in UCB [19,20,21,22], few studies were conducted on the RIs for presepsin, KL-6, and SP-A [23,24,25,26]. Therefore, we aimed to establish the RIs for presepsin, KL-6, and SP-A in large-scaled samples and to evaluate the possible influence of neonatal and maternal factors on presepsin, KL-6, and SP-A levels in UCB.

## 2. Materials and Methods

### 2.1. Study Population and Sample Collection

This retrospective study was conducted at the Konkuk University Medical Center (KUMC), a 900-bed tertiary-care hospital. From April 2020 to May 2022, UCB samples were collected from umbilical cord veins using syringes when neonates were delivered. This study consecutively obtained 613 UCB samples without the selection of the maternal population. It was difficult to obtain a sufficient number of samples from healthy neonatal subjects and it provoked ethnic issues related to the use of neonatal samples [27]. Thus, an indirect method for determining the RIs was used [28]. Samples were collected into a serum separating tube (Greiner BioOne GmbH, Frickenhausen, Germany) and were immediately centrifuged at 2300 rpm for 10 min. The separated sera were frozen at −70 °C in small aliquots to avoid repeated freezing and thawing until testing. Presepsin was claimed to be stable at −20 °C or lower for up to 60 days by a manufacturer, and KL-6 and SP-A were not reported for stability. Samples were excluded if they (1) were hemolytic and clotted samples and (2) did not have sufficient volume for testing. Maternal and neonatal demographic and clinical information were collected using electronic medical records: neonatal sex, single or twin, gestational age (GA), birth weight (BW), Apgar score at one and five minutes (min), maternal age, delivery mode, the presence of premature rupture of membranes (PROM), gestational diabetes mellitus (GDM), and pre-eclampsia. The characteristics of the study population are summarized in Table 1. This study defined preterm as a neonate born before 37 gestational weeks. Low BW was defined as weight less than 2500 g. The levels of Apgar scores were divided into two groups based on seven, since seven to ten Apgar scores at one and five min are considered normal. The healthy group was defined as a group with neonatal factors with the term, normal BW, and above 7 Apgar score and maternal factors with NSVD and without PROM, GDM, and pre-eclampsia. The Institutional Review Board of the KUMC reviewed this study protocol and exempted the approval of the study with waived informed consent (KUMC, 2020-07-027).

### 2.2. Measurement of Presepsin, KL-6, and SP-A Level

Presepsin, KL-6, and SP-A levels were measured in 613 UCB samples according to manufacturers’ instructions. The frozen samples were thawed at room temperature for at least 30 min. The presepsin, KL-6, and SP-A levels were measured using HISCL Presepsin assay kit, HISCL KL-6 assay kit, and HISCL SP-A assay kit (Sysmex Corp., Kobe, Japan) on a fully automated analyzer, the HISCL-5000 (Sysmex Corp., Hyogo, Japan), in batches within 2 months of storage. These kits were developed based on the sandwich chemiluminescence enzyme immunoassay. Each assay utilized two types of mouse monoclonal antibodies against presepsin, KL-6, or SP-A in the samples: biotinylated antibodies and the alkaline phosphatase (ALP)-labeled antibodies. ALP decomposed substrate to an intermediate, which emitted a luminescent signal. The intensity of the emitted signal was calculated according to the calibration curve. A quality control assay was performed daily after calibration at two levels according to the manufacturer’s instructions. The coefficient of variation value of within-run, between-run, or within-laboratory precision at two levels showed <11.5% for presepsin, <2.2% for KL-6, and <1.8% for SP-A. The analytical measurement ranges were 20 to 30,000 pg/mL for presepsin, 10 to 6000 U/mL for KL-6, and 1.0 to 1000 ng/mL for SP-A.

### 2.3. Statistical Analysis

Data were checked for normal distribution and homogenous variation by the Kolmogorov–Smirnov test. Data were presented as numbers (percentage) or medians (interquartile range, IQR). In a total of 613 samples, extreme outliers (37 in presepsin, 12 in KL-6, and 4 in SP-A) were excluded by a double-sided Grubbs test, and the remaining samples (576 in presepsin, 601 in KL-6, and 609 in SP-A) were analyzed. For presepsin, KL-6, and SP-A, a non-parametric method was used to establish the RIs according to the Clinical Laboratory Standards Institute (CLSI) guidelines EP28-A3C [29]. The study population was divided into subgroups according to neonatal factors, including neonatal sex, 37 gestational weeks, BW of 2500 g, seven Apgar scores at one min, and maternal factors, including delivery mode, the presence of PROM, GDM, and pre-eclampsia. Previous studies on factors affecting presepsin, KL-6, or SP-A levels showed controversial results, so common factors used in neonatal and maternal studies were selected for partitioning [23,24,25,26]. The robust method was performed for the subgroup with small samples, less than 120, to compare the RIs between subgroups. In the robust method, the bootstrapping estimated the confidence intervals for RIs, randomly creating large data from the initial results. The median and IQR of the subgroup were compared by the Mann–Whitney U test. The following formula calculated the statistical significance of the mean difference between subgroups: *z* = $\left|x_{1}-x_{2}\right|/{\left[\left(\frac{s_{1}{}^{2}}{n_{1}}\right)+\left(\frac{s_{2}{}^{2}}{n_{2}}\right)\right]}^{1/2}$, where $x_{1}$ and $x_{2}$ are the means of each subgroup, where $s_{1}\;\mathrm{and}\;s_{2}$ are the variances, and where $n_{1}\;\mathrm{and}\;n_{2}$ are the numbers of total reference numbers in each subgroup [24]. The critical value, z^*^, was determined to be 4.8 by the formula of z^*^ = $3{\left[(n_{1}+n_{2})/240\right]}^{1/2}$ [29]. Age-related RIs were estimated by z-scores normal distribution analysis. All statistical analyses used MedCalc Software (version 20.014, MedCalc Software, Ostend, Belgium).

## 3. Results

Presepsin, KL-6, and SP-A levels showed non-parametric distributions and left-skewed histograms (*p* < 0.001 for all) (Figure 1). In the total group, the median levels for presepsin, KL-6, and SP-A were 186.5 pg/mL (range 28–508 pg/mL), 76.0 U/mL (range 20–213 U/mL), and 16.1 ng/mL (range 0.2–61.0 ng/mL), respectively (Table 2). The RIs of presepsin, KL-6, and SP-A levels were 64.9–428.3 pg/mL, 43.0–172.0 U/mL, and 2.1–36.1 ng/mL, respectively. Table 2 also shows the RIs of presepsin, KL-6, and SP-A levels for subgroups divided by neonatal factors. The median levels for subgroups by neonatal factors did not show statistical differences. The z-values for all neonatal factors were less than the critical value (z^*^ = 4.8). The RIs of presepsin, KL-6, and SP-A levels for subgroups divided by maternal factors were presented in Table 3. The median levels for subgroups by maternal factors did not show a statistical difference. The z-values for all maternal factors were less than the critical value (z^*^ = 4.8). Therefore, a single RI could apply to neonatal and maternal factors for presepsin, KL-6, and SP-A. In the healthy group, the RIs of presepsin, KL-6, and SP-A levels were 62.0–458.0 pg/mL, 35.1–154.7 U/mL, and 2.5–38.5 ng/mL, respectively (Table 4). In presepsin and SP-A, the z-values for sex were less than the critical value (z^*^ = 4.8). In KL-6, the z-value for sex was 8.5, which is higher than the critical value (z^*^ = 4.8) (*p* = 0.015).

Regarding the associations of presepsin, KL-6, and SP-A levels with GA, the presepsin level for 34 to 40 gestational weeks and SP-A level for all gestational weeks showed relatively constant 2.5th and 97.5th centile RIs (Figure 2). The ranges from 2.5th to 97.5th centile Ris of KL-6 levels were 53.1–77.5, 36.3–110.2, 26.8–132.4, 24.3–143.6, and 28.6–143.7 for 24, 28, 32, 36, and 40 gestational weeks. The median level and 97.5th centile RI of KL-6 showed a slight increase as the GA increased.

## 4. Discussion and Conclusions

This study aimed to establish RIs for presepsin, KL-6, and SP-A levels in UCB. We compared the RIs according to neonatal and maternal factors; there was no statistical difference between subgroups according to neonatal sex, 37 gestational weeks, a BW of 2500 g, seven Apgar scores at one min, delivery mode, PROM, GDM, and pre-eclampsia. The median level and 97.5th centile RI of KL-6 showed a slight increase with the increase in GA, but not presepsin and SP-A.

Establishing RIs in neonates is challenging. First, a prevalent portion of neonate samples can be collected from hospitalized patients, and it is very difficult to obtain healthy neonate samples that serve as a reference. Second, regarding the weight of the neonate, it is difficult to obtain a sufficient volume of samples, and blood collection in preterm neonates and low birth weight neonates is more restricted than in healthy neonates [30]. Third, various alterations would occur during the transition from the fetal to the neonatal period [31]. Last, the neonate is a rather heterogeneous group with a wider range of GA than that of pediatrics and adult. Therefore, it is not surprising that there are few studies about neonatal RIs, and neonatologists still struggle to diagnose their patients. Nevertheless, establishing RIs is essential in neonates.

For presepsin, this study showed that the median levels and RIs were 186.5 pg/mL (range 28–508 pg/mL) and 64.9–428.3 pg/mL in the total group and 192.0 pg/mL (range 28–497 pg/mL) and 62.0–458.0 pg/mL in the healthy group. In a previous study, the median levels and RIs were 620 pg/mL and 352–1370 pg/mL in the preterm group and 603.5 pg/mL and 315–1178 pg/mL in the term group, respectively [25]. The previous study used a direct method with whole blood (capillary blood via heel puncture), whereas this study used an indirect method with UCB. The direct comparison of RIs was difficult for both studies using different methodologies. However, this study showed remarkably low RIs. Another study reported physiologic variation in the early neonatal period. The median levels were 318.5 pg/mL (range 99.2–1180 pg/mL) at birth, 343.8 pg/mL (range 129.0–655.0 pg/mL) at first day of life, and 180.5 pg/mL (range 89.5–421.5 pg/mL) at fifth day of life [32]. In those previous studies, the RIs were measured using whole blood, whereas this study measured RI using UCB. Depending on the blood origin, such as UCB, whole blood, or venous blood, the results could show significantly different consequences [33]. Thus, it is essential to establish presepsin RIs in UCB independently. In this study, RIs between subgroups divided by neonatal and maternal factors did not show a statistical difference. A previous study compared presepsin RIs in UCB, term, and preterm neonates related to neonatal sex, GA, delivery mode, BW, PROM, pre-eclampsia, GDM, and neonatal clinical status [26]. Similar to this study, RIs in UCB did not show statistical differences except PROM. The previous study included PROM with maternal chorioamnionitis, whereas this study included PROM without any maternal infection sign. Amniotic fluid and UCB are normally sterile environments from antigenically abundant exposure. Antigen exposure in the intrauterine environment, such as chorioamnionitis, may indicate a remarkable increase in presepsin levels [34]. In addition, our findings revealed that presepsin RIs were independent of GA, although studies on the association between presepsin RI and GA have been previously reported controversially. The design of this study using UCB differed from the previously reported studies that measured presepsin levels using whole blood after birth and analyzed GA by a binary comparison of preterm and term [10,35]. 

Our findings revealed that the median levels and RIs of KL-6 were 76.0 U/mL and 43.0–172.0 U/mL in the total group and 77.0 U/mL and 35.1–154.7 U/mL in the healthy group. There were no neonatal and maternal factors affecting KL-6. In agreement with this study, only one study reported that the median level of KL-6 was 73.0 U/mL and RI was 44.3–148.2 U/mL in UCB. The distributions between KL-6 levels with sex, GA, BW, delivery mode, and Apgar scores at one min were not statistically significant [24]. In this study, the RIs between the total group and the healthy group did not show a significant difference, whereas the RIs between males and females in the healthy group showed statistical significance. However, the clinical significance is uncertain. Thus, further investigation would be necessary for the effect of KL-6 on sex. The median level and 97.5th centile RI of KL-6 tended to be slightly increased as GA increased. KL-6 is mainly produced by type II alveolar epithelial cells [13]. The injury of alveolar epithelial cells increased alveolar vascular permeability, and KL-6 levels in blood increased [36]. KL-6 levels were higher in children and adults than in neonates because of increased exposure to environmental risk factors such as air pollution and smoking [24]. However, UCB used in this study is not yet affected by pulmonary respiration, so alveolar epithelial cell injury cannot be explained. The results of this study could present that alveolar or bronchial maturation according to GA increase could contribute to the increase in KL-6 levels.

SP-A has a potential role in innate immune responses by facilitating the pulmonary clearance of bacterial or viral infection [37]. In addition, SP-A is associated with acute lung injury and lung fibrosis, indicating alveolar damage and leakage of SP-A into the circulation [38]. In this study, the median levels and RIs were 16.1 ng/mL and 2.1–36.1 ng/mL in the total group and 15.7 ng/mL and 2.5–38.5 ng/mL in the healthy group. The RIs of SP-A between subgroups according to neonatal and maternal factors did not show statistical differences, and SP-A levels indicated constant levels according to the increase in GA. Similar to this study, a normal range of SP-A was previously reported from 3.5 to 20.7 ng/mL in UCB [23]. However, in disagreement with this study, the previous study revealed the association between the SP-A level and GA (*p* = 0.0001) and between the SP-A level and BW (*P* = 0.002). However, the previous study explained that the increased SP-A levels were related to the labor effect caused by uterine contraction rather than GA or BW [23]. Pressure such as labor promoted the secretion of pulmonary surfactant and increased SP-A levels in alveolar spaces [39]. Elevated pressure sequentially contributed to the leakage of SP-A from alveolar spaces into the blood [23]. Contrary to the previous study, this study was not analyzed for factors that could induce SP-A leakage. Therefore, we concluded that there was no association between SP-A and GA.

This study has several limitations. First, although UCB is a safe and non-invasive method to measure parameters in neonates, it is difficult to directly reflect or predict the neonate status after birth because the transition from the fetus to the neonate is the most complex physiologic process that occurs in many organs and blood [30]. Thus, this study can better predict the intrauterine status of neonates rather than the postnatal period. Second, this study compared the distributions of RIs for presepsin, KL-6, and SP-A according to neonatal and maternal factors. We did not explore the multidimensional aspects of multiple factors that could affect the level of each biomarker. However, this study aimed to establish RIs for presepsin, KL-6, and SP-A levels and to compare distributions of presepsin, KL-6, and SP-A levels between each subgroup.

In conclusion, this study established the RIs of presepsin, KL-6, and SP-A levels in UCB according to the CLSI guidelines (EP28-A3C). The RIs of presepsin, KL-6, and SP-A levels were not affected by neonatal and maternal factors, so single RIs could be applied to each factor for the three biomarkers. The median level and 97.5 centile RI of KL-6 tended to be slightly increased as GA increased, and it should be noted that the variation in KL-6 level might increase as lung maturation progresses. Further investigation would be needed to determine the clinical significance.

## Figures and Tables

**Figure 1 diagnostics-12-02213-f001:**
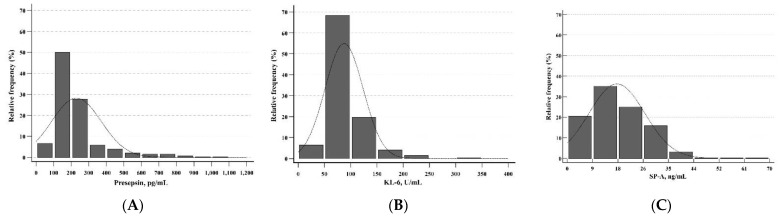
Distributions of presepsin, KL-6, and SP-A in UCB (*n* = 613). The distributions of (**A**) presepsin, (**B**) KL-6, and (**C**) SP-A were all non-parametric and showed left-skewed histograms. Abbreviations: KL-6, Krebs von den Lungen 6; SP-A, surfactant protein A; UCB, umbilical cord blood.

**Figure 2 diagnostics-12-02213-f002:**
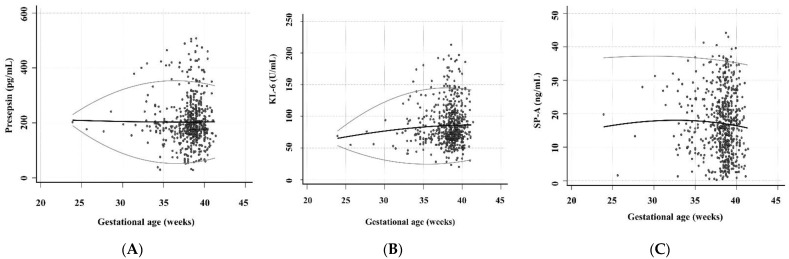
The levels of presepsin, KL-6, and SP-A according to the GA (*n* = 576 in presepsin, *n* = 601 in KL-6, and *n* = 609 in SP-A). These scatter diagrams show the levels of (**A**) presepsin, (**B**) KL-6, and (**C**) SP-A according to the GA. The black central line indicated the median level and the gray curved line indicated 2.5th to 97.5th centile RIs. Abbreviations: KL-6, Krebs von den Lungen 6; SP-A, surfactant protein A; UCB, umbilical cord blood; GA, gestational age.

**Table 1 diagnostics-12-02213-t001:** Characteristics of the study population (*n* = 613).

Characteristics	UCB
Neonate information	
Male, *n* (%)	329 (53.7)
Twin, *n* (%)	40 (6.5)
GA, weeks	38.6 (37.7–39.3)
Preterm, *n* (%)	106 (17.3)
BW, g	3145 (2790–3450)
Low BW, *n* (%)	93 (15.2)
Apgar score at 1 min	8 (7–8)
Apgar score at 5 min	9 (9–9)
Maternal information	
Age, years	34.0 (34.0–34.0)
C/S, *n* (%)	328 (53.5)
PROM, *n* (%)	85 (13.9)
GDM, *n* (%)	93 (15.2)
Pre-eclampsia, *n* (%)	36 (5.9)
Marker	
Presepsin, pg/mL	191 (160–248)
KL-6, U/mL	76.5 (64.5–99.8)
SP-A, ng/mL	16.2 (10.1–23.4)

Abbreviations: UCB, umbilical cord blood; *n*, number; GA, gestational age; BW, birth weight; min, minute; C/S, cesarean delivery; PROM, premature rupture of membranes; GDM, gestational diabetes mellitus; KL-6, Krebs von den Lungen 6; SP-A, surfactant protein A.

**Table 2 diagnostics-12-02213-t002:** Reference intervals of the total group for presepsin, KL-6, and SP-A according to neonatal factors.

Parameters	Total	Sex		GA		BW		Apgar Score at 1 min	
		Male	Female	Term	Preterm	Normal	Low	≥7	<7
Presepsin, pg/mL									
*n*	576	310	266	480	96	488	88	485	91
Median (range)	186.5 (28–508)	186.0 (28–508)	187.5 (38–489)	186.0 (28–508)	187.5 (29–465)	185.0 (28–508)	190.5 (29–465)	188.0 (28–506)	183.0 (38–508)
RI	64.9–428.3	67.8–436.1	59.7–424.7	64.1–432.0	5.3–361.2	66.2–431.6	13.6–361.3	66.1–448.7	2.5–354.0
Lower limit ^a^	59.0–77.0	31.0–83.0	54.0–82.0	59.0–81.0	2.3–38.2	60.0–77.0	−16.7–47.5	56.0–85.0	−12.7–17.7
Upper limit ^a^	409.0–460.0	393.0–497.0	404.0–460.0	394.0–461.0	330.2–389.5	403.0–461.0	327.4–390.9	403.0–382.8	319.2–387.4
Z-value		0.8		0.6		0.6		0.9	
KL-6, U/mL									
*n*	601	323	278	498	103	509	92	507	94
Median (range)	76.0 (20–213)	76.0 (23–197)	78.0 (20–213)	77.0 (20–213)	74.0 (28–192)	77.0 (20–213)	74.0 (28–181)	76.0 (20–213)	78.0 (28–189)
RI	43.0–172.0	39.2–157.8	45.0–186.0	44.0–171.5	9.8–141.1	45.0–172.8	6.9–142.3	43.7–172.6	15.3–146.5
Lower limit ^a^	34.0–46.0	31.0–46.0	34.0–48.0	34.0–46.0	−0.7–21.7	34.0–47.0	−2.4–18.5	34.0–47.0	3.5–27.2
Upper limit ^a^	156.0–186.0	147.0–175.0	163.0–191.0	158.0–187.0	127.6–153.2	156.0–189.0	128.7–154.6	156.0–187.0	134.0–158.3
Z-value		1.6		0.4		0.5		0.4	
SP-A, ng/mL									
*n*	609	327	282	504	105	516	93	512	97
Median (range)	16.1 (0.2–61.0)	15.7 (0.2–43.0)	16.8 (0.5–61.0)	16.1 (0.2–61.0)	16.2 (0.5–41.3)	16.0 (0.2–61.0)	16.8 (0.5–36.9)	16.3 (0.2–61.0)	15.7 (0.5–37.1)
RI	2.1–36.1	2.4–35.8	1.6–36.8	2.4–36.3	1.8–36.1	2.4–36.5	1.3–36.1	2.5–36.3	3.5–37.0
Lower limit ^a^	1.3–2.6	1.3–2.7	0.7–2.6	1.2–2.6	0.3–3.3	1.2–2.6	−0.3–2.9	1.5–2.7	1.3–6.2
Upper limit ^a^	34.8–37.2	34.7–37.1	34.6–40.5	34.7–37.3	33.2–38.7	34.9–39.4	33.4–38.6	34.3–36.6	33.5–39.9
Z-value		0.3		0.6		0.5		0.1	

^a^ Lower limit and upper limit were represented as the 90% confidence interval of reference limits. Abbreviations: KL-6, Krebs von den Lungen 6; SP-A, surfactant protein A; GA, gestational age; BW, birth weight; min, minute; *n*, number; RI, reference range.

**Table 3 diagnostics-12-02213-t003:** Reference intervals of the total group for presepsin, KL-6, and SP-A according to maternal factors.

	Delivery Mode		Presence of PROM		Presence of GDM		Presence of Pre-Eclampsia	
	NSVD	C/S	Non-PROM	PROM	Non-GDM	GDM	Non-Pre-Eclampsia	Pre-Eclampsia
Presepsin, pg/mL								
*n*	272	304	495	81	490	86	544	32
Median (range)	189.5 (28–497)	202.3 (38–508)	188.0 (28–506)	179.0 (60–508)	186.5 (28–508)	186.0 (51–489)	186.5 (28–508)	187.0 (38–409)
RI	60.0–430.4	71.1–428.6	63.6–431.2	6.0–342.4	69.7–428.9	16.1–349.7	65.3–430.8	0.1–355.4
Lower limit ^a^	31.0–82.0	56.0–81.0	55.0–81.0	−26.9–43.6	60.0–83.0	−16.9–51.2	59.0–79.0	−39.7–53.3
Upper limit ^a^	360.0–461.0	410.0–489.0	409.0–461.0	306.1–379.5	410.0–460.0	317.1–381.7	410.0–461.0	304.6–412.6
Z-value	0.1		1.3		1.1		0.2	
KL-6, U/mL								
*n*	281	320	517	84	513	88	566	35
Median (range)	75.0 (20–213)	77.5 (74–83)	77.0 (20–213)	74.0 (45–192)	76 (23–213)	78.5 (20–192)	77 (20–213)	73 (28–135)
RI	34.3–171.8	44.0–174.9	41.0–171.1	10.4–133.7	48.9–172.3	12.3–146.3	45.0–173.7	21.0–122.4
Lower limit ^a^	30.0–46.0	39.0–47.0	33.0–45.0	−2.0–25.3	35.0–47.0	1.1–26.5	34.0–47.0	9.6–35.5
Upper limit ^a^	147.0–192.0	151.0–187.0	156.0–187.0	119.4–148.7	156.0–186.0	132.9–159.2	158.0–187.0	109.4–136.0
Z-value	1.4		1.0		0.3		2.3	
SP-A, ng/mL								
*n*	284	325	525	84	518	91	573	36
Median (range)	16.3 (0.2–61.0)	15.9 (0.5–44.2)	16.0 (0.5–61.0)	16.7 (0.2–41.3)	15.9 (0.2–61.0)	17.6 (0.5–37.2)	16.1 (0.2–61.0)	15.9 (1.5–35.6)
RI	2.3–37.2	1.6–35.5	2.3–35.9	3.2–37.8	2.4–36.5	0.9–35.0	2.1–36.4	3.1–37.2
Lower limit ^a^	0.7–2.7	1.2–2.6	1.3–2.6	0.5–6.1	1.3–2.6	−0.7–2.5	1.2–2.6	0.5–2.6
Upper limit ^a^	33.4–41.3	34.6–36.9	34.5–37.1	34.1–41.2	34.9–39.4	32.5–37.3	34.7–37.3	31.7–41.4
Z-value	0.5		0.8		0.0		0.2	

^a^ Lower limit and upper limit were represented as the 90% confidence interval of reference limits. Abbreviations: KL-6, Krebs von den Lungen 6; SP-A, surfactant protein A; NSVD, normal spontaneous vaginal delivery; C/S, Cesarean delivery; PROM, premature rupture of membranes; GDM, gestational diabetes mellitus; *n*, number; RI, reference range.

**Table 4 diagnostics-12-02213-t004:** Reference intervals of the healthy group for presepsin, KL-6, and SP-A.

Parameters	Total	Sex	
		Male	Female
Presepsin, pg/mL			
*n*	167	91	76
Median (range)	192.0 (28.0–497.0)	188.0 (28.0–497.0)	195.5 (60.0–481.0)
RI	62.0–458.0	15.9–363.3	36.3–351.9
Lower limit ^a^	28.0–102.0	−16.3–52.8	1.7–73.1
Upper limit ^a^	403.0–497.0	325.8–395.1	313.8–385.2
Z-value		1.5	
KL-6, U/mL			
*n*	171	91	80
Median (range)	77.0 (23.0–172.0)	71.0 (23.0–172.0)	83.0 (40.0–167.0)
RI	35.1–154.7	16.0–129.8	24.3–138.0
Lower limit ^a^	23.0–48.0	7.1–27.8	14.6–33.9
Upper limit ^a^	144.0–172.0	118.7–139.2	127.1–150.0
Z-value		8.5	
SP-A, ng/mL			
*n*	609	327	282
Median (range)	15.7 (1.3–61.0)	15.7 (0.2–43.0)	16.8 (0.5–61.0)
RI	2.5–38.5	−3.7–32.9	−3.0–36.4
Lower limit ^a^	1.3–3.6	−6.3–1.2	−7.6–1.2
Upper limit ^a^	33.0–61.0	29.8–36.0	32.1–41.0
Z-value		2.2	

^a^ Lower limit and upper limit were represented as the 90% confidence interval of reference limits. Abbreviations: see Table 2.

## Data Availability

The data presented in this study are available from the corresponding author upon reasonable request.

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
