# Peer review of "Distribution of Presepsin, Krebs von den Lungen 6, and Surfactant Protein A in Umbilical Cord Blood"

_diagnostics, 2022, doi:10.3390/diagnostics12092213_

Round 1

Reviewer 1 Report

Diagnostics-1865516-peer-review-v1

This is a study of local importance, where the reference interval was determined by indirect approach (indirect method) for presepsin, Krebs von den Lungen-6 and Surfactant Protein-A in umbilical cord blood, the manuscript presents several methodological inaccuracies that should be resolved.

Notes for the authors

Abstract: Correlation is a measure of the linear relationship (covariance) between two continuous quantitative variables (x, y).

Rather, the purpose of the study is to associate a quantitative variable with qualitative variables such as the gender of the newborn, mode of delivery, etc. I suggest you change the term from correlation to association and only use it when you want to test the association between two continuous quantitative variables.

Introduction: The objective proposed in the summary of the manuscript is different from the one proposed at the end of the introduction, in the first one it refers to the correlation of the concentrations of the three biomarkers analyzed and the maternal and neonatal variables and in the one mentioned later there is no reference to the correlations.

Methods: It is essential that you describe the method for determining the reference intervals (direct or indirect), as well as justify why one method was used and not the other.

Methods: What strategy was used to ensure that the analytical methods and the population of newborns remained stable during the period of collection of samples and information from newborns and mothers?

Methods: What was the accuracy and precision of each of the tests?

Also, what was the inter-assay or intra-assay variability?

Methods: Why is the analysis of association (correlation) not described in the statistical analysis? If association (correlation) analysis is part of your goal.

Methods: The different variables that were considered for partitioning (division into subgroups) were selected based on the known effects on presepsin, KL-6 and SP-A or were random.

Results: It is difficult to assess the quality of a manuscript when figures and tables are not available. For example, I cannot know what the number of reference neonate samples is for each partition, in order to assess whether the reference interval is statistically reliable for each subgroup.

Discussion: It is not possible to compare their reference interval with the reference interval published by Pugni et al., since a different methodology was used to obtain the interval (one is direct and the other indirect).

The study by Pugni et al. It was based on the collection of samples of newborns from a preselected reference population, to carry out the measurements and then determine the intervals.

In this study, you used an alternative approach to analyze the results generated as part of routine pathology tests and performed statistical techniques to determine reference intervals.

Discussion: It is important to mention that the main limitation of the present study is the possible effects of the diseased subpopulations in the derived interval, since it was not taken into account whether or not the samples of the newborns were from healthy or sick newborns at birth.

Conclusion: When you conclude, it is important that you describe the importance of the reference interval and clinical decision limits, as reference intervals are generally considered a distribution of test values in the predefined population and should not be confused with clinical decision limits. Which are determined primarily through patient assessment as a result of response or changes in management.

Author Response

Response to reviewers’ comments

Diagnostics-1865516-R

Distribution of Presepsin, Krebs von den Lungen 6, and Surfactant Protein A in Umbilical Cord Blood

We really appreciate the reviewers and editor for the time and effort on our manuscript. During the revision, we recognized the weakness of the manuscript and could have a chance to improve its quality. We tried to reflect your valuable comments in the revised manuscript and hope our effort would be satisfactory to you.

Reviewer #1:

This is a study of local importance, where the reference interval was determined by an indirect approach (indirect method) for presepsin, Krebs von den Lungen-6 and Surfactant Protein-A in umbilical cord blood, the manuscript presents several methodological inaccuracies that should be resolved.

  1. Abstract:Correlation is a measure of the linear relationship (covariance) between two continuous quantitative variables (x, y). Rather, the purpose of the study is to associate a quantitative variable with qualitative variables such as the gender of the newborn, mode of delivery, etc. I suggest you change the term from correlation to association and only use it when you want to test the association between two continuous quantitative variables.

Thank you for your comment. According to your comment, we modified the following sentences in Abstract section and specified the purpose of the study in detail.

This study aimed to establish reference intervals (RIs) of presepsin, KL-6, and SP-A levels and to evaluate the possible influence of neonatal and maternal factors on presepsin, KL-6, and SP-A levels in umbilical cord blood (UCB). (page 1, line 23)

  1. Introduction:The objective proposed in the summary of the manuscript is different from the one proposed at the end of the introduction, in the first one it refers to the correlation of the concentrations of the three biomarkers analyzed and the maternal and neonatal variables and in the one mentioned later there is no reference to the correlations.

Thank you for your comment. According to your comment, we modified sentences in Abstract section and Introduction section and specified the purpose of the study in detail.

This study aimed to establish reference intervals (RIs) of presepsin, KL-6, and SP-A levels and to evaluate the possible influence of neonatal and maternal factors on presepsin, KL-6, and SP-A levels in umbilical cord blood (UCB). (page 1, line 23)

Therefore, we aimed to establish the RIs for presepsin, KL-6, and SP-A in large-scaled samples and to evaluate the possible influence of neonatal and maternal factors on presepsin, KL-6, and SP-A levels in UCB. (page 2, line 74)

  1. Methods:It is essential that you describe the method for determining the reference intervals (direct or indirect), as well as justify why one method was used and not the other.

Thank you for your comment. According to your comment, we added the following sentences in Material and methods section. We also updated references in Introduction section and added references in Materials and methods section.

It was difficult to obtain a sufficient number of samples from healthy neonatal subjects and it provokes ethnic issues related to the use of neonatal samples [27]. Thus, an indirect method for determining the RIs was used [28]. (page 2, line 84)

  1. Maldeghem, I.; Nusman, D.M.; Visser,D.H. Soluble CD14 subtype (sCD11-ST) as biomarker in neonatal early-onset sepsis and late-onset sepsis: a systemic review and meta-analysis. BMC Immunol 2019, 20, 17, doi:10.1186/s12865-019-0298-8. (page 11, line 352)
  2. Poggi, C.; Lucenteforte, E.; Petri, D.; Masi, S.D.; Dani, C. Presepsin for the diagnosis of neonatal early-onset sepsis: a systematic review and meta-analysis. JAMA Pediatr 2022, 176, 750-758, doi:10.1001/jamapediatrics.2022.1647. (page 11, line 354)

  1. Kang, T.; Yoo, J.; Jekarl, D.W.; Chae, H.; Kim, M.; Park, Y. J.; Oh, E. J.; Kim Y. Indirect method for estimation of reference intervals of inflammatory markers. Ann Lab Med 2023, 43, 55-63, doi.org/10.3343/alm.2023.43.1.55. (page 11, line 388)
  2. Jones, G.R.D.; Haeckel, R.; Loh, T.P.; Sikaris, K.; Streichert, T.; Katayev, A.; Barth, J.H.; Ozarda, Y.; behalf of the IFCC Committee on Reference Intervals and Decision Limits. Indirect methods for reference interval determination – review and recommendations. Clin Chem Lab Med 2018, 19, 57, 20-29. doi:10.1515/cclm-108-0073. (page 11, line 3901)

  1. Methods:What strategy was used to ensure that the analytical methods and the population of newborns remained stable during the period of collection of samples and information from newborns and mothers?

 Thank you for your comment. We modified the following sentences in Material and method section.

This retrospective study was conducted at the Konkuk University Medical Center (KUMC), a 900-bed tertiary-care hospital. From April 2020 to May 2022, UCB samples were collected from umbilical cord veins using syringes when neonates were delivered. This study consecutively obtained 613 UCB samples without the selection of the maternal population. (page 2, line 80)

The separated sera were frozen at -70°C in small aliquots to avoid repeated freezing and thawing until testing. Presepsin was claimed to be stable at -20°C or lower for up to 60 days by a manufacturer, and KL-6 and SP-A have not been reported for stability. (page 2, line 88)

The presepsin, KL-6, and SP-A levels were measured using HISCL Presepsin assay kit, HISCL KL-6 assay kit, and HISCL SP-A assay kit (Sysmex Corp., Kobe, Japan) on a fully automated analyzer, the HISCL-5000 (Sysmex Corp., Hyogo, Japan) in batches within 2 months of storage. (page 4, line 114)

A quality control assay was performed daily after calibration at two levels according to the manufacturer’s instructions. (page 4, line 122)

  1. Methods: What was the accuracy and precision of each of the tests? Also, what was the inter-assay or intra-assay variability?

 Thank you for your comment. According to your comment, we added the following sentence in Material and method section.

The coefficient of variation value of within-run, between-run, or within-laboratory precision at two levels showed <11.5% for precepsin, <2.2% for KL-6, and <1.8% for SP-A. (page 4, line 123)

  1. Methods:Why is the analysis of association (correlation) not described in the statistical analysis? If association (correlation) analysis is part of your goal.

 Thank you for your comment. We evaluated the possible influence of neonatal and maternal factors for presepsin, KL-6, and KL-6. Thus, we compared the distribution of each subgroup for presepsin, KL-6, and SP-A.

The robust method was performed for the subgroup with small samples, less than 120, to compare the RIs between subgroups. (page 5, line 141)

  1. Methods:The different variables that were considered for partitioning (division into subgroups) were selected based on the known effects on presepsin, KL-6, and SP-A or were random.

Thank you for your comment. According to your comment, we added the following sentences in Material and method section. We modified the references.

Previous studies on factors affecting presepsin, KL-6, or SP-A levels showed controversial results, so common factors used in neonatal and maternal studies were selected for partitioning [23-26]. (page 5, line 138)

  1. Cho, K.; Matsuda, T.; Okajima, S.; Matsumoto, Y.; Sagawa, T.; Fujimoto, S.; Kobayashi, K. Factors influencing pulmonary surfactant protein A levels in cord blood, maternal blood and amniotic fluid. Biol Neonate 1999, 75, 104-110. (page 12, line 378)
  2. Uchida, Y.; Minowa, H.; Ebisu, R.; Nishikubo, T.; Takahashi, Y.; Yoshioka, A. Normal values for KL-6 in cord venous plasma of neonates. Pediatr Int 2007, 49, 167-171, doi:10.1111/j.1442-200X.2007.02345.x. (page 12, line 380)
  3. Pugni, L.; Pietrasanta, C.; Milani, S.; Vener, C.; Ronchi, A.; Falbo, M.; Arghittu, M.; Mosca, F. Presepsin (soluble CD14 subtype): reference ranges of a new sepsis marker in term and preterm neonates. PLoS One 2015, 10, e0146020, doi:10.1371/journal.pone.0146020. (page 12, line 382)
  4. Ergor, S.N.; Yalaz, M.; Koroglu, O.A.; Sozmen, E.; Akisu, M.; Kultursay, N. Reference ranges of presepsin (soluble CD14 subtype) in term and preterm neonates without infection, in relation to gestational and postnatal age, in the first 28 days of life. Clin Biochem 2020, 77, 7-13, doi:10.1016/j.clinbiochem.2019.12.007. (page 12, line 385)

  1. Results: It is difficult to assess the quality of a manuscript when figures and tables are not available. For example, I cannot know what the number of reference neonate samples is for each partition, in order to assess whether the reference interval is statistically reliable for each subgroup.

 Thank you for your comment. We added the following sentences in Material and method section and added the number of samples in the figures and tables.

In a total of 613 samples, extreme outliers (37 in presepsin, 12 in KL-6, and 4 in SP-A) were excluded by double-sided Grubbs test, and the remaining samples (576 in presepsin, 601 in KL-6, and 609 in SP-A) were analyzed. (page 4, line 131)

Figure 1. Distribution of presepsin, KL-6, and SP-A in UCB (N = 613). (page 5, line 170)

Figure 2. The levels of presepsin, KL-6, and SP-A according to the GA (N = 576 in presepsin, N = 601 in KL-6, and N = 609 in SP-A). (page 9, line 218)

Table 2. Reference intervals of the total group for presepsin, KL-6, and SP-A according to neonatal factors. (page 6, line 174)

Table 3. Reference intervals of the total group for presepsin, KL-6, and SP-A according to maternal factors. (page 7, line 178)

  1. Discussion: It is not possible to compare their reference interval with the reference interval published by Pugni et al., since a different methodology was used to obtain the interval (one is direct and the other indirect). The study by Pugni et al. It was based on the collection of samples of newborns from a preselected reference population, to carry out the measurements and then determine the intervals. In this study, you used an alternative approach to analyze the results generated as part of routine pathology tests and performed statistical techniques to determine reference intervals.

 Thank you for your comment. To reflect your comment, we added the following sentences in Discussion section. Please accept our modification and explanation.

For presepsin, this study showed that the median levels and RIs were 186.5 pg/mL (range 28 – 508 pg/mL) and 64.9 – 428.3 pg/mL in the total group and 192.0 pg/mL (range 28 – 497 pg/mL) and 62.0 – 458.0 in the healthy group. In a previous study, the median levels and RIs were 620 pg/mL and 352 – 1,370 pg/mL in pre-term group and 603.5 pg/mL and 315 – 1,178 pg/mL in term group, respectively [25]. The previous study used a direct method with whole blood (capillary blood via heel puncture), whereas this study used an indirect method with UCB. Direct comparison of RIs was difficult for both studies using different methodologies. However, this study showed remarkably low RIs. (page 9, line 240)

  1. Discussion:It is important to mention that the main limitation of the present study is the possible effects of the diseased subpopulations in the derived interval, since it was not taken into account whether or not the samples of the newborns were from healthy or sick newborns at birth.

Thank you for your comment, and we totally agree with you on the main limitation. To reflect your comment, we added the following table in Result section and added related sentences in Materials and method and Discussion sections.

Table 4. Reference intervals of the healthy group for presepsin, KL-6, and SP-A. (page 8, line 183)

Parameters

Total

Sex

Male

Female

Presepsin, pg/mL

 N

167

91

76

 Median (range)

192.0 (28.0 – 497.0)

188.0 (28.0 –497.0)

195.5 (60.0 – 481.0)

 RI

62.0 – 458.0

15.9 – 363.3

36.3 – 351.9

 Lower limita

28.0 – 102.0

-16.3 – 52.8

1.7 – 73.1

 Upper limita

403.0 – 497.0

325.8 – 395.1

313.8 – 385.2

 Z-value

1.5

KL-6, U/mL

 N

171

91

80

 Median (range)

77.0 (23.0 – 172.0)

71.0 (23.0 – 172.0)

83.0 (40.0 – 167.0)

 RI

35.1 – 154.7

16.0 – 129.8

24.3 – 138.0

 Lower limita

23.0 – 48.0

7.1 – 27.8

14.6 – 33.9

 Upper limita

144.0 – 172.0

118.7 – 139.2

127.1 – 150.0

 Z-value

8.5

SP-A, ng/mL

 N

609

327

282

 Median (range)

15.7 (1.3 – 61.0)

15.7 (0.2 – 43.0)

16.8 (0.5 – 61.0)

 RI

2.5 – 38.5

-3.7 – 32.9

-3.0 – 36.4

 Lower limita

1.3 – 3.6

-6.3 – -1.2

-7.6 – 1.2

 Upper limita

33.0 – 61.0

29.8 – 36.0

32.1 – 41.0

 Z-value

2.2

aLower limit and upper limit were represented as the 90% confidence interval of reference limits. Abbreviations: see table 2.

The healthy group was defined as a group with neonatal factors with the term, normal BW, and above 7 of APGAR score and maternal factors with NSVD and without PROM, GDM, and preeclampsia. (page 2, line 100)

In the healthy group, the RIs of presepsin, KL-6, and SP-A levels were 62.0 – 458.0 pg/mL, 35.1 – 154.7 U/mL, and 2.5 – 38.5 ng/mL, respectively (Table 4). In presepsin and SP-A, the z-values for sex were less than the critical value (z* = 4.8). In KL-6, the z-value for sex was 8.5, which is higher than the critical value (z* = 4.8) (P = 0.015). (page 5, line 165)

In this study, RIs between the total group and the healthy group did not show a significant difference, whereas RIs between sex in the healthy group showed statistical significance. However, the clinical significance is uncertain. Thus, further investigation would be necessary for the effect of KL-6 on sex. (page 10, line 273)

  1. Conclusion: When you conclude, it is important that you describe the importance of the reference interval and clinical decision limits, as reference intervals are generally considered a distribution of test values in the predefined population and should not be confused with clinical decision limits. Which are determined primarily through patient assessment as a result of response or changes in management.

Thank you for your comment. According to your comment, we modified the following sentences in Conclusion section.

Further investigation would be needed to determine the clinical significance. (page 11, line 318)

Reviewer 2 Report

Distribution of Presepsin, Krebs von den Lungen 6, and Surfactant Protein A in Umbilical Cord Blood

By Minjeong Nam, et al.

The Authors show three markers studied in the neonatal umbilical cord. Presently the studied markers do not accomplish some significant diagnostic usefulness. The study seems more suitable as academic investigation in neonatal physiological changes. Since this paper seems rightly written it is suitable for publication. The data are useful for comparison in future investigations.

Author Response

Thank you very much for your review.

Round 2

Reviewer 1 Report

The manuscript has improved considerably, as well as each of the observations made have been answered, so I consider that the manuscript currently meets the criteria for publication.

Notes for the authors

I only have two new observations

I suggest that the authors check the spelling of the manuscript.

It is suggested that the authors adhere to the recommendations of the International System of Units